# Algorithmic Graph Theory, Reinforcement Learning and Game Theory in MD Simulations: From 3D Structures to Topological 2D-Molecular Graphs (2D-MolGraphs) and Vice Versa

**DOI:** 10.3390/molecules28072892

**Published:** 2023-03-23

**Authors:** Sana Bougueroua, Marie Bricage, Ylène Aboulfath, Dominique Barth, Marie-Pierre Gaigeot

**Affiliations:** 1Université Paris-Saclay, University Evry, CY Cergy Paris Université, CNRS, LAMBE UMR8587, 91025 Evry-Courcouronnes, France; 2Université Paris-Saclay, University Versailles Saint Quentin, DAVID, 78000 Versailles, France

**Keywords:** algorithmic graph theory, molecular dynamics, identification of conformers, machine learning, game theory, pathways, conformational conversion, prediction of 3D structures

## Abstract

This paper reviews graph-theory-based methods that were recently developed in our group for post-processing molecular dynamics trajectories. We show that the use of algorithmic graph theory not only provides a direct and fast methodology to identify conformers sampled over time but also allows to follow the interconversions between the conformers through graphs of transitions in time. Examples of gas phase molecules and inhomogeneous aqueous solid interfaces are presented to demonstrate the power of topological 2D graphs and their versatility for post-processing molecular dynamics trajectories. An even more complex challenge is to predict 3D structures from topological 2D graphs. Our first attempts to tackle such a challenge are presented with the development of game theory and reinforcement learning methods for predicting the 3D structure of a gas-phase peptide.

## 1. Introduction

In the last decade, the domains of artificial intelligence (AI) and algorithmic graph theory have seen a wealth of development in the realm of molecular simulations. On the one hand, the large domain of machine learning (ML) has seen a growth in popularity in the community of molecular dynamics (MD) simulations. A wealth of development has, in particular, been dedicated to the machine learning of force fields (FF) for realizing more accurate classical MD simulations. Pioneers and well-advanced research groups in ML-FF include the groups of Behler, Czanyi, Shapeev and Ceriotti; see for instance [1,2,3,4,5,6,7,8,9,10], with applications mostly done in the condensed phase. Marquetand et al. [11] pioneered machine learning for infrared spectroscopy calculation, by coupling ML-FF with the simultaneous machine learning of dipole moments that uses similar training as that for energies and forces in ML-FF. Other works such as in [4,12,13] continued on the same routes for IR and Raman spectroscopies with various types of observables being learned. In the context of vibrational spectroscopy based on MD simulations, we developed a theoretical route based on atomic polar tensors (APTs) and Raman tensors, applicable to the modeling of IR, Raman and SFG (Sum Frequency Generation) spectroscopies; see [14,15,16,17]. The APT and Raman tensors can be machine learned from high level quantum calculations. Vibrational spectra can also be directly machine learned in order to predict very quickly a spectrum associated to a given molecular system; see e.g., [18,19,20,21].

On the other hand, algorithmic graph theory, a branch of AI, is powerful for the analysis of molecular simulations. A 2D graph encodes topological properties of matter through vertices and edges that report on the specific (pairwise) interactions between the vertices. At the molecular level of representation, the vertices are usually associated to atoms while the edges report on the interactions between the atoms, e.g., chemical bonds and intermolecular interactions. Three-dimensional information related to e.g., distances, angles, are not encoded into topological 2D graphs unless vertices/nodes are specifically colored with such information (giving rise to 3D graphs). Modeling the interaction between the atoms of a molecule by a 2D graph is nowadays well established [22], as already shown by the early work of James Joseph Sylvester (1814–1897). Easier to obtain or to predict than 3D graphs, 2D graphs already carry information on the structure, the functional properties and even the 3D shape of the molecules they model. Examples include the classification of similar molecules according to their topology [23,24], the prediction of patterns in biological molecules [25], the prediction of the 3D structure of small molecules [26], etc. Molecular graphs are also commonly used in supervised machine learning algorithms; the framework of graph-based models for molecules is indeed naturally suited for carrying out predictions in message-passing neural network schemes.

In bio-informatics and chemo-informatics, the challenge is to identify or design algorithms capable of obtaining molecular properties from input graphs and to follow these properties in time. This can be done by operations research or artificial intelligence approaches, considering each graph as an input [27], or by machine learning approaches, considering all atoms or sub-graphs of any target molecular graph as interconnected agents [26,28]. Obtaining, analyzing and qualifying such algorithms frequently gives rise to new research challenges in algorithmics; see e.g., [29].

The last decade has seen several developments on algorithmic graph theory for analyzing various types of molecular dynamics simulations, from e.g., the conformational analysis of gas phase molecules and clusters, to their chemical reactivity, to the dynamics of H-Bonds in liquids, to the dynamics of the solvation shells of ions in liquids, to the structural and dynamical analysis of complex aqueous interfaces in condensed matter; see for instance [30,31,32,33,34,35,36,37,38,39,40,41].

Our group has been involved in the development of graph-theory-based methods for analyzing (post-processing) atomistic molecular dynamics simulations (obtained from DFT-based MD and classical FF-MD). These developments are included in the GaTewAY post-processing software described in [42]. We present our developed algorithmic graph theory methods in Section 2. The applications are presented in the same section, showing the versatility of the graphs and associated algorithms in going from the analysis of MD trajectories of ‘simple’ gas phase molecules to much more complex heterogeneous aqueous interfaces in condensed matter. It is one thing to be able to define and follow in time topological 2D graphs from an input molecular dynamics simulation; an even more complex challenge is to be able to make the reverse processing, i.e., to predict the 3D structure of a molecular system from the sole knowledge of its 2D topological graph. Section 3 of this paper reports our first attempts to tackle this challenge by implementing game theory and reinforcement learning. An illustration on the prediction of the 3D structure of a gas phase flexible peptide using a 2D molecular graph as input is presented. Conclusions and perspectives are presented in Section 4.

## 2. GaTewAY Post-Processing Tool for Time-Dependent Conformational Analysis

We have developed 2D molecular graphs (labeled 2D-MolGraph) and associated algorithms in order to automatically analyze the conformational dynamics of molecules and their assemblies from molecular dynamics simulations. The 2D-MolGraph and the algorithms are described in the next section. Different applications are presented in order to illustrate the versatility and transferability of the 2D-MolGraphs that can be applied to gas phase molecules, clusters, liquids, solids and inhomogeneous interfaces between solids and liquids. Our developments are included in the GaTewAY (Graph Theory for conformAtional dYnamics analysis) software described in [42].

A key component in the versatility and transferability of our 2D-MolGraphs is the atomic level of granularity that was chosen to define the vertices and edges of topological 2D graphs. In our 2D-MolGraphs, any molecular conformation is defined by a molecular graph where vertices represent the atoms of the molecular system and the edges represent the interactions between these atoms. These interactions can result in covalent bonds, hydrogen bonds, these interactions can be of electrostatic nature, etc. As will be seen in Section 2.1, one can easily define and implement new relevant interactions that are needed to describe a given molecular system and hence augment the number of definitions for the edges in the 2D-MolGraphs.

With 2D-MolGraphs in our hands, the exploration with time of molecular conformations along molecular dynamics simulations can easily be seen as the exploration of graph topologies, which can be tracked using graph-theory-based methods, e.g., isomorphism. This is described in Section 2.2. Given an MD trajectory, the GaTewAY software thus provides the list of conformations that have been explored based on the isomorphism test from [43], their residence time and the “graph of transitions” that represents all transitions that have been observed between the identified conformations along the MD time-length. Examples of applications to gas phase molecules and to more complex inhomogeneous aqueous interfaces will be given in Section 2.3. GaTewAY can be applied to any kind of MD trajectory, be it *ab initio* MD, classical MD or coarse-grained MD, or any type of mixing between these atomistic representations, including chemical reactivity. The algorithms are implemented in such a way that they can be easily extended to take into account any other types of interactions and parameters to analyze the conformational dynamics.

### 2.1. From the 3D Structure to a 2D MolGraph

We summarize here the key elements to define a 2D MolGraph; more details are found in [40,42]. A molecular conformation is defined by a **mixed graph** $G=(V,E_{C},A_{H},E_{I},E_{O})$, where the following definitions are applied (these are the current definitions implemented in GaTewAY; these definitions can easily be extended as previously emphasized):*V* is the set of all **atoms** of the molecular system except hydrogen atoms that are not accounted for. Each atom of the molecular system is a **vertex** of *G*.$E_{C}=\{\left[a,b\right],a\in V,b\in V:\left[a,b\right]$ is a **covalent bond**}, where each covalent bond represents an **edge** in *G* ([a,b]=[b,a]).$A_{H}=\{\left(a,b\right),a\in V,b\in V:\left(a,b\right)$ is an **H-bond**}, where each H-bond represents an **arc** (directed edge) in *G* ((a,b)≠(b,a)), in which atom *a* is a heavy atom in the H-bond and atom *b* is the acceptor of the hydrogen bond. Directed edges (arcs) are hence used to keep the information related to the hydrogen atom involved in H-bonds: *for a hydrogen bond, the edge is directed from the heavy atom to the donor.*There is therefore no need to include explicitly the hydrogen atoms in the 2D MolGraphs, only the ones involved in H-bonds are indirectly incorporated through the directed edges. There are no other directed edges in the 2D MolGraphs.$E_{I}=\{\left[a,b\right],a\in V,b\in V:\left[a,b\right]$ is an **“electrostatic interaction”**}, where each electrostatic interaction represents an **edge** in *G* ([a,b]=[b,a]). Such interaction can be obtained whenever an ion (anion or cation) atom interacts with other atoms.$E_{O}=\{\left[a,b\right],a\in V,b\in V:\left[a,b\right]$ is an **“organometallic interaction”**}, where each organometallic interaction represents an **edge** in *G* ([a,b]=[b,a]). Such interaction is defined whenever a metallic atom such as Manganese, Ruthenium or Gold (metallic atoms investigated in our works up-to-now) interacts with any other type of atom.

In order to take into account the chemical type of the atoms in 2D MolGraphs, we apply a special case of graph coloring, such that the vertices of a given 2D MolGraph display the same color *if and only if* the corresponding atoms have the same chemical type. In Figure 1, the vertices of the 2D MolGraph (right side) are colored in such a way that dark gray stands for carbon atoms, blue stands for nitrogen atoms, red for oxygen atoms, and light gray for hydrogen atoms. Moreover, the edges in the 2D MolGraphs are colored in order to distinguish the bond type. For example, edges in black lines stand for covalent bonds and red dashed arcs are for hydrogen bonds. These are conventions that are systematically applied to 2D MolGraphs.

The right side of Figure 1 shows the 2D MolGraph that is associated to a given 3D conformation of a gas phase peptide (shown on the left side of the figure). It illustrates the concepts of vertices and edges of a given 2D MolGraph as defined above, and it illustrates the coloring of the vertices and edges just described.

### 2.2. Fingerprinting Based on Bonding Patterns and Types of Interactions

To analyze the conformational dynamics of a molecular system either for one trajectory or multiple trajectories treated in parallel, our software GaTewAY is composed of two main stages:**Construction of the series of 2D MolGraphs:** This stage consists in constructing the series of 2D MolGraphs from the list of the positions of the atoms given by the trajectory file(s). At each step (snapshot) *I* of the MD trajectory, the list of atoms and their cartesian coordinates are known. Depending on the chemical type, we have established a database in which each possible atom has a specific covalent radius and a maximum number of covalent bonds that it can form. For each type of bond/interaction, a set of threshold distance/angle values have been established. These values are in a database that can be easily modified by any user. We always choose the nearest atoms to fix the eventual bond overflow, especially when the user sets high cut-off values. For example, given an atom *a* that should form at most one covalent bond, and two atoms *b* and *c* that respect the condition to form a covalent bond with atom a, whenever the distance(a,b) is less than distance(a,c), the algorithm will choose atom *b* for establishing a covalent bond from atom *a* as atom *b* has the shortest distance from atom *a*. In order to reduce the computational costs of the geometric analysis at each step of the trajectory, two essential features have been implemented: one feature is **the orbit of an atom** which keeps track of the subset of atoms that can potentially form bonds with atom *a*; the second feature is composed of **reference snapshots** that are a (small) subset of snapshots where the orbits are recalculated. For all details, we refer the reader to [40].**Graph theory analysis:** Once orbits and bonds are computed using the geometric-based approach, the structural and dynamical behaviors of the molecular system are examined with graph-theoretic methods. The 2D MolGraph allows defining a unique system topology (i.e., both intra- and inter-molecular bonds) for any possible configuration of atoms along the trajectory. Any conformational change, such as isomerization, proton transfer, change in solvation shells, etc., can be seen as the exploration of a different 2D MolGraph topology.

Graph isomorphism, as defined in [43], allows to represent each conformer with a fingerprint graph and allows comparisons between graphs. In our applications, isomorphism consists in comparing the distribution of edges between two 2D MolGraphs: if the two graphs being compared have the same set of bonds/interactions connected to the same set of atoms (in terms of chemical types/colors for the graphs), these graphs are then isomorphic, i.e., the two graphs are identical; see [43,44,45,46].

In a formal way, we define the isomorphism between two conformations as follows: two conformations $G_{i}=\left(V_{i},E_{C_{i}},A_{H_{i}},E_{I_{i}}\right)$ and $G_{j}=\left(V_{j},E_{C_{j}},A_{H_{j}},E_{I_{j}}\right)$ are **isomorphic** if and only if there exists a bijection $\theta_{i,j}$:$V_{i}\to V_{j}$ such as:$\forall a\in V_{i}$, we have $\phi \left(a\right)=\phi (\theta_{i,j}\left(a\right))$.$\left[a,b\right]\in E_{C_{i}}\Leftrightarrow \left[\theta_{i,j}\left(a\right),\theta_{i,j}\left(b\right)\right]\in E_{C_{j}}$.$\left(a,b\right)\in A_{H_{i}}\Leftrightarrow \left(\theta_{i,j}\left(a\right),\theta_{i,j}\left(b\right)\right)\in A_{H_{j}}$.$\left[a,b\right]\in E_{I_{i}}\Leftrightarrow \left[\theta_{i,j}\left(a\right),\theta_{i,j}\left(b\right)\right]\in E_{I_{j}}$.

Using the graph coloring based on the chemical type of atoms, we only allow the permutation between vertices of the same color. In Figure 2, the graphs in the lower row show the coloring of graphs (A) and (B). Here, vertex *N* (shown in blue) in graph (C) and vertex *O* (shown in red) in graph (D) do not have the same color; hence, graphs (C) and (D) are not isomorphic.

The main steps of the GaTewAY algorithm are as follows:Initialization: read the first snapshot $I_{1}$ of the trajectory and construct the first 2D-MolGraph $G_{1}$ (see definitions above).Read a new snapshot $I_{i}$ and construct the associated 2D-MolGraph $G_{i}$.Test if $G_{i}$ is isomorphic to $G_{i-1}$. If they are isomorphic, add the snapshot $I_{i}$ to the list of appearance of $G_{i-1}$ and go to step (2).Test if $G_{i}$ is isomorphic to one of the 2D-MolGraph already identified. If yes, then add the snapshot $I_{i}$ to the list of appearance of the isomorphic 2D-MolGraph and go to step (2).Else, add $G_{i}$ to the list of the 2D-MolGraph already identified.Return to step (2) in order to read the subsequent snapshot.

The conformational changes that occur over the molecular dynamics can also be summarized as a graph called **“graph of transitions”**. An example is given in Figure 3. Given trajectory I composed of *S* snapshots, the graph of transitions $G_{I}=\left(G,A\right)$ is labeled as a directed graph, wherein:$G=\{G_{1},G_{2},\ldots ,G_{C}\}$ are the vertices of $G_{I}$. Each vertex represents a 2D-MolGraph (i.e., a conformation) that has been identified at least once over trajectory I.A are the arcs (directed edges) of $G_{I}$. Each arc $\left(G_{i},G_{j}\right)$ represents a transition observed from conformation $G_{i}$ to conformation $G_{j}$.

For a given vertex in the graph of transitions, we report the percentage of appearance of this conformation in the trajectory as well as the bond(s)/interaction(s) that were formed in this conformation. Figure 3 shows an example of a graph of transitions. This graph is composed of six vertices; each vertex represents one conformer that has been identified along the trajectory. We can observe that conformer (3) is the most frequent with a total percentage of appearance of 45.57%. The edges between the vertices are also labeled, providing two types of information: (1) the total frequency rate for going from one conformation to the other one, (2) the bonds/interactions that have changed when going from one conformation to the other. The colors of the vertices in the graph of transitions directly give the most relevant conformations in terms of appearance periods. We hence put in red the conformations that appear at least $P_{min}\%$ (input parameter that the user can change), and the ones in the green color are those occurring below this threshold. All the conformations explored along the molecular dynamics simulations can be kept in the graph of transitions. Such information might indeed be useful for some analyses, typically when rare events (rare conformations) are investigated. The user can modify this at will. For instance, one can observe a larger conformational dynamic between conformations (2) and (3) in the graph of transitions in Figure 3; such event occurs 60 times. One can see on the graph that the hydrogen bond N1−O1 appears in going from conformation 2 to conformation 3.

### 2.3. Evaluation and Validation

The GaTewAY software can analyze any molecular dynamics trajectory, i.e., AIMD, QM-MM-MD, FF-MD, CG-MD. Because of the atomic granularity level chosen for the 2D-MolGraph definitions, GaTewAY is versatile and can be applied to trajectories of gas phase molecules and clusters, to solids, nano-particles, to any complex inhomogeneous molecular system as e.g., solid–water interfaces. In the following, we illustrate applications of the 2D-MolGraphs and associated graph algorithms in the gas phase (analysis of the trajectory of an isolated protonated peptide $Z-Ala_{6}-COOH,C_{26}H_{39}N_{7}O_{8}$) and in the condensed phase for trajectories of hydrophobic surfaces in contact with liquid water (air/water, graphene/water, boron-nitrate BN/water).

#### 2.3.1. Time-Dependent Structural Recognition of Peptides

Our first illustration of the capability of 2D-MolGraphs and associated graph algorithms concerning the trajectory of the gas phase $Z-Ala_{6}-COOH$ peptide ($C_{26}H_{39}N_{7}O_{8}$, 80 atoms). A 150 K DFT-MD trajectory composed of 43,125 snapshots is analyzed (∼20 ps, δt=0.5 fs). Based on the analysis of the H-bonds dynamics, 46 different conformers of the $Z-Ala_{6}-COOH$ peptide have been identified by GaTewAY over the trajectory, showing a very dynamical and flexible peptide in the gas phase, even at a rather moderate temperature. At the temperature of 150 K, eight different types of hydrogen bonds have been explored in $Z-Ala_{6}-COOH$ over the trajectory. The structure of each identified conformer is composed of two to six hydrogen bonds formed simultaneously. The peptide is found either in opened structures where a low number of simultaneous H-bonds are present or in H-bonded folded structures.

Figure 4 presents two 2D-MolGraphs corresponding to two identified conformers of $Z-Ala_{6}-COOH$ (see the top of the Figure 4), respectively built on five (Figure 4c) and four H-bonds (Figure 4d). Three of these H-bonds are common in the two conformations, while the two structures differ by the N2⋯O8 vs. N1⋯O8 H-bond. Figure 5 presents the graph of transitions that summarizes the dynamics of the $Z-Ala_{6}-COOH$ gas phase peptide at 150 K. It is composed of an extremely high number of vertices and edges connecting these vertices, which is the signature of high dynamicity and high flexibility of the peptide.

The high number of vertices and edges in the graph of transitions in Figure 5 illustrates the limit of the atomic level of granularity used in the 2D-MolGraphs for highly flexible molecular systems. There are also brief periods of time for breaking/reforming covalent bonds and/or H-bonds over the dynamics that are captured as changes in conformations by the graph analysis, as they are signatures of dynamics of these bonds around the threshold values employed in the method. These numerous transitions between conformers are in practice the signature of the existence of a single ‘meta-conformation’, around which dynamicity and flexibility occur.

One advantage of topological 2D graphs is that higher levels of granularity can be employed for the graphs. Because the 2D-MolGraphs are especially based on the H-bond dynamics, one way to achieve a more coarse-grained analysis for flexible peptides such as $Z-Ala_{6}-COOH$ is to define graphs based on the H-bonded rings/cycles and their polymorphism, i.e., the similar role an H-bonded ring will play in two different conformations despite the H-bonded cycle being built on different atoms in the peptide. The underlying paradigm here is that the structure of such H-bonded molecule is mainly related to the interactions (i.e., sharing of edges and/or vertices) between the cycles of the associated 2D-MolGraph. We are currently developing such graph approach. To describe the method briefly, using an evolution of the Horton algorithm, we consider a basis made of H-bonded cycles in each conformation of a given trajectory, i.e., a subset of cycles of minimum size allowing by composition to generate all the cycles of the conformation. Two cycles in the obtained bases of two conformations are equivalent if they share at least some essential atoms, and if they have the same interactions in their respective conformations with other cycles also pairwise equivalent. Given a trajectory (seen again as a sequence of graphs having the same set of vertices), a maximum set of pairwise equivalent cycles (also called “polymorphic” cycles) is called a poly-cycle. With this in hand, one can build the polygraph of the trajectory, which is the graph in which a vertex is one identified poly-cycle. There is an edge between two vertices, i.e., between two poly-cycles, if they have an interaction in all conformations containing one polymorphic cycle of each of these two poly-cycles. Thus, each molecular conformation can be characterized by a subgraph of this polygraph (depending of the basis of its cycles), and the evolution of these subgraphs over the trajectory represents the evolution of the structure of the molecule, without the “noisy” modification of links within each cycle discussed above at the molecular level of representation. The trajectory can therefore be seen as the interaction of the cycles evolving over time in their atomic structure (polymorphism) and still interacting in the same way, but with some of these cycles appearing or disappearing over time.

The first experimentation and analysis of such innovative algorithmic approach is currently being implemented in our group. As an example, Figure 6 shows the 2D-MolGraphs of two different conformers of $Z-Ala_{6}-COOH$. In the two 2D-graphs of the peptide, the H-bonded rings have been highlighted in different colors, where each color corresponds to a given polymorphic H-bonded ring. In this example, there is one hydrogen bonded cycle (in orange in Figure 6) that is different between the two conformers: these two (orange) H-bonded rings involve different atoms and give rise to a different size of the cycle while they are polymorphic to each other. They hence play the same role within the associated 3D structures. Despite the two H-bonded cycles being built on different atoms, the associated coarse-grained graph of cycles that is found (and reported at the bottom of Figure 6) is the same for these two 2D-MolGraphs because of the similar role played by the two (orange) H-bonded cycles in the two conformations. These two different roles appear in the orange vertex of the graph of cycles where there are two identities for the H-Bond reported for this particular cycle, while the other vertices of the graph of cycles have one single identity.

#### 2.3.2. The 2D-MolGraphs in Condensed Matter: Hydrophobicity Revealed at Aqueous Interfaces

The 2D-MolGraphs and associated graph algorithms can be applied to more complex and challenging molecular systems such as inhomogeneous aqueous interfaces in the condensed phase. Here we present algorithmic graph theory applied to the DFT-MD dynamics of three hydrophobic aqueous interfaces, which have been characterized at the molecular level for their hydrophobic character in [47]. We have shown in [41,47,48] that the surface hydrophobicity results in the formation of a two-dimensional (2D) highly collective H-bond network made by the water molecules in the layer in direct contact with the hydrophobic surface (i.e., water located in the BIL-Binding Interfacial Layer as defined in [17]), where the water–water H-bonds are formed parallel to the surface. This water collective 2D-Hbonded-Network is the molecular signature of surface hydrophobicity.

We here illustrate the specific characteristics of the 2D-MolGraph of the 2D-HBonded-Network and the algorithms that have been used in order to dissect further the 2D-MolGraph, and provide more details about the actual structure of the water molecules in this collective H-bonded network. In particular, the actual organization of the water molecules in the 2D-Network, i.e., the network topology, is shown to vary between the hydrophobic surfaces. Three hydrophobic aqueous interfaces are here dissected, i.e., the air/liquid water interface as the prototype of hydrophobic surfaces, the graphene/liquid water interface and the Boron-Nitrate BN/liquid water interface. DFT-MD trajectories are used (see their analyses in [47]); a limited number of 400 snapshots have been extracted and analyzed with graph theory for each trajectory (from a total of 50 ps trajectory per system). This corresponds to roughly an analysis of 1 snapshot every 0.1 ps of dynamics, which represents a good statistical sampling regarding the dynamics of hydrogen bonds. For each aqueous interface, the graph analyses are done on the BIL-interfacial region only, in which there is roughly an average of 48 water molecules (all simulation boxes are roughly equivalent in sizes).

Figure 7 presents one 2D-MolGraph per aqueous interface, from left to right: the BIL organization of the air/water interface, the graphene/water interface and the BN/water interface. For the illustrations, we present one 2D-MolGraph per system; there are as many 2D-MolGraphs as there are non-isomorphic graphs along each trajectory. The statistical analysis of the ensemble of 2D-MolGraphs per interface is presented in the next paragraph by employing algorithmic graphs.

Each of these 2D-MolGraphs shows the same global structural property of the water molecules in the BIL of hydrophobic surfaces: a collective arrangement of the water molecules in terms of H-bonded polygons (or rings) that are adjacent to each other.

To obtain the statistical view on the number of water molecules that are interconnected within the 2D-HBonded-Network, the (identified non-isomorphic) 2D-MolGraphs can be analyzed in terms of the size of the connected components, i.e., the ensemble of subgraphs in which all vertices are connected to each other. This is done for the 400 2D-MolGraphs that have been calculated in each trajectory. The distribution of the connected components is shown in Figure 8 for each aqueous interface. One can immediately see that the three plots are similar, with the largest extended network made by ∼90–95% of the water molecules. From the 2D-MolGraphs in Figure 7 and from the distributions of connected components in Figure 8, one can thus conclude that the water molecules in the BIL are statistically organized with the same collective HB-Network at all three interfaces, displaying the same high degree of inter-connectivity.

The supplementary information that can be directly deduced from the analysis of the 2D-MolGraphs is the distribution of the size of the H-bonded polygons made by the water molecules in the BIL at the interface with the three hydrophobic surfaces investigated here. This is obtained by applying the Horton algorithm [49] that provides a minimum basis of cycles for one given graph. Such analysis provides a direct view on the graph topology and thus on the structural topology of the water molecules at the interface. The final analyses are reported in Figure 9 for the three investigated hydrophobic aqueous interfaces.

Very interestingly, though the water molecules in the three investigated BILs are assembled with the same collective 2D-Hbonded-Network, the distribution of sizes of the H-bonded polygons that build these networks are non-identical between the three hydrophobic interfaces. On the one hand, the sizes of the H-bonded polygons are centered on 4 to 6 for the air/water and BN/water interfaces. There is a clear dominant component related to H-bonded pentagons made by the water molecules at the interface with the BN surface while the formation of H-bonded tetragons and pentagons is found equivalent at the interface with the air. On the other hand, the 2D-Hbonded-Network made by the water molecules at the surface of graphene is more homogeneous in terms of sizes of the polygons, where tetragons, hexagons and heptagons have roughly the same probability of appearance, and H-bonded pentagons dominate slightly more. We hence see that the water molecules dominantly form five-membered H-bonded rings/polygons at the interface with the BN surface, which can be associated to the hexagonal templated structure of Boron Nitride. The length of the C−C covalent bonds in BN is shorter than the O⋯H hydrogen bonds: the best arrangement for the water molecules is thus into five-membered H-bonded rings rather than six-membered rings.

One can also calculate the average percentage of water molecules that belong to the H-bonded rings/polygons that build the 2D-Hbonded-Network at the direct interface with the solid. This network is composed of 2D polygons that can be adjacent to each other but also of chains of water molecules that can make links between some of the polygons. We hence find that the percentage of water molecules that give rise to the polygons within the 2D-HBonded-Network is around 40–30% for the aqueous interfaces investigated here, with the following interesting ranking: one finds a larger percentage of water in the 2D polygons for the air/water interface (∼43.44%) than for the graphene/water interface (∼36.16%) interface, which has a larger percentage than for the BN/water interface (∼33.57%). Such percentages of participation of the water molecules might explain the strength of the 2D-HBonded-Network found at each interface. The work in [47] indeed shows that the strength of the 2D-HBonded-Network can be ranked as Air>Graphene>BN. In other words, the more water molecules forming rings within the 2D-Hbonded-Network (i.e., the more rings being formed), the stronger the 2D-HBonded-Network, and the more hydrophobic the interface.

## 3. Game Theory and Reinforcement-Learning to Predict the 3D Structures of Gas Phase Molecules

The previous section has shown how we can use graph theory to go from 3D structures to 2D molecular graphs and a few usages of these graphs together with graph algorithms for the direct analysis of molecular dynamics simulations. The next challenge that we address in this section concerns the reverse route/back-mapping, i.e., how to go from a given 2D-MolGraph to the associated 3D structure of the molecule. Such reverse route is typically necessary in the theoretical domain of conformer generation that is needed for e.g., starting the search of low energy conformers in quantum mechanics calculations, for starting molecular dynamics simulations by sampling the phase space, and in computational drug design and docking. Methods for generating the 3D structures of conformers have been developed in the literature. See for instance [50,51,52,53,54,55,56]. Some of these methods typically rely on databases of 3D structures of sub-units that will be used to generate a reasonable conformation of a more complex molecule. Other methods rely on crystallographic databases for the initial construction. Nowadays, topological databases based on 2D graphs such as the ones presented in the first part of this paper are becoming more in use. See for instance the topology-based databases of [18,57]. Furthermore, the 3D prediction of e.g., proteins typically takes advantage of the large amount of available 3D data and the specificity of local structures in these biomolecules [58] to propose prediction approaches by Deep Learning (see e.g., [59]), which turns out to be interesting alternatives to energy minimization approaches for these particular molecules.

In our specific context, the present development is part of this latter domain, i.e., to generate 3D structures of molecular conformers from a database of 2D graphs. In our case, the database is built on the 2D-MolGraphs that were presented in Section 2 of this paper. We show in this section of the paper the algorithmic methods that we have developed based on game theory and reinforcement learning in order to generate the 3D structure of a gas phase molecule from the sole knowledge of its 2D-MolGraph. We stick to isolated molecules in the present presentation and application. The methodology should be transferable to the more complex domain of the condensed phase. This is, however, non-trivial; we are currently working on this aspect together with improvements of the methodology for gas phase molecules, as will be discussed later at the end of this section.

Many popular prediction methods have to be supervised by the knowledge of the real (or confirmed) three-dimensional structure of the molecules; see for instance [50,60]. These methods are system-dependent. Unsupervised approaches have been developed based on heuristic or meta-heuristic techniques; see for example [27,51].

We show in this section some of our preliminary works using game theory algorithms associated with a reinforcement-learning approach, inspired by methods developed by some of us in bio-informatics for the prediction of the 3D structure of RNA biopolymers at a high granularity level of representation, where the 3D structure is made of stems and loops; see [26,61,62,63]. In these previous works, as in the present work, we assume that the 3D structure of a molecule is based on the notion of equilibrium between local components. The leading idea is to apply stochastic learning methods to find the molecule’s equilibrium space corresponding to stable or meta-stable situations. Such approach is adapted here at the atomic level of representation by considering only the topological constraints imposed by the molecular graph and the local constraints imposed on the atoms by their neighborhood. The key point is that energy is never introduced in the method, which makes the method independent on the accuracy of the level of representation of the energy. The method is hence typically not affected by the never-ending discussions of quality of quantum mechanics methods (high level accuracy such as CCSDT types of methods vs. lower level of accuracy such as the DFT) vs. much lower level of energy accuracy represented by parameterized force fields representations. Furthermore, as a key element in finding 3D structures at the lowest possible computational cost, our method does not depend on the computational cost of the evaluation of energy (and forces).

In the following, we describe the key components of the developed methods and show our first preliminary result in an application to the prediction of the 3D structure of a tri-peptide molecule in the gas phase from the sole knowledge of its 2D-MolGraph (as defined in Section 2 of this paper).

### 3.1. Game Model

We consider a 2D-MolGraph G=(V,E) as defined in Section 2 of this work, where *V* is the set of vertices (representing *all* the atoms of the molecular system, i.e., including all hydrogen atoms added to the definition presented in Section 2) and *E* is the set of possible edges (made of covalent bonds and hydrogen bonds for the peptidic molecules of interest here). The goal is to generate the 3D configuration associated to a given 2D-MolGraph by convergence towards a chemically realistic configuration. The approach we propose consists in representing any edge in a given 2D-MolGraph, i.e., representing any bond (u,v) (hereby covalent bonds and hydrogen bonds) as a player in a sequential game (see [64]), whose objective is to find a comfortable position with respect to its neighbors while minimizing as much as possible the negative impact of its choices on the equilibrium of the whole molecular structure. We consider a sequential game where players (u,v) choose a 3D position in space in a certain order. We repeat the game in consecutive rounds. At the end of each round of the game, each player is assigned a utility ranking calculated from a utility function that takes into account both the local equilibrium and the impact of each player upon the equilibrium of the other players who will be playing next. This utility function will influence the choice of this player in the next round of the game, by using a reinforcement learning method (as used in [26]) that makes the players’ strategies converge towards those that induce an equilibrium situation for all the players. It is the definition of the utility functions that must guarantee that this equilibrium is close to a chemically realistic situation.

At each round *k* of the game, the vertices of *G* are ordered from 1 to n=|V| from running a Breadth-First Search (BFS) Algorithm, as given in [65], in order to browse the graph *G* and construct a covering tree *T*. When the $i^{th}$ player associated to *u* is active, we consider each (u,v) couple with $v\in N_{T}\left(u\right)$ being a child of *u* in *T*. The couple (u,v) is then a player in the game. Possible strategies of (u,v) are the possible positions of *v* in a discretized 3D space of icosahedrons centered on *u*, denoted I(u), with 42 possible atomic positions located on it. The position of the root of *T* is (0,0,0). Each player (u,v) thus has 42 possible strategies. The chosen strategy provides a virtual space position of *v* in I(u), relatively to the one of *u* chosen by player (w,u), with *w* the parent of *u* in *T*. A rotation is applied to I(u) around *u*, such that the virtual space of the parent *w* of *u* in I(u) previously fixed matches with the real final position of *w*. The rotation allows for minimizing the average displacement of each point of I(u). After applying this rotation, each vertex of $N_{T}\left(u\right)$ has its definitive position in the game’s current iteration.

Consider that at the end of the 2(n−1) steps of the ongoing round of the game (i.e., the number of (u,v) couples), each vertex *u* of the 2D-MolGraph has a position $p^{k}\left(u\right)=\left(x_{u}^{k},y_{u}^{k},z_{u}^{k}\right)$ in the global space. The utility ranking of vertex *u* is computed from the two utility functions $Direct^{k}\left(u\right)$ and $Indirect^{k}\left(u\right)$:$Direct^{k}\left(u\right)$: considers on the one hand all the relative positions of the vertices *v* in $N_{T}\left(u\right)$ in I(u), and on the other hand for any $w\in N_{G}\left(u\right)-N_{T}\left(u\right)$, the projection of $p_{k}\left(w\right)$ on I(u). $Direct^{k}\left(u\right)$ is hence the minimum sum of the distances separating these positions from the ideal positions that the neighbors should have around the atom if they respect the valence shell electron pair repulsion theory (VSEPR). VSEPR is used here for the geometrical rules known for chemical groups. For instance, a CH${}_{4}$ group should have a tetrahedral symmetry in space, etc.$Indirect^{k}\left(u\right)$: given any vertex *u*, we first consider the sum for all vertices $v\in N_{G}\left(u\right)-N_{T}\left(u\right)$ of the euclidian distances between $proj_{w}\left(v\right)$, the projection of $p_{k}\left(v\right)$ on I(u), and $p_{k}\left(v\right)$, the position of *v* in the round *k* of the game. Then for all vertices *z* in *G* such that $p^{k}\left(z\right)$ is in I(w), we also add the sum of the euclidian distance between $p^{k}\left(z\right)$ and $p^{k}\left(u\right)$. Let us denote by $local^{k}\left(w\right)$ the global sum of these distances. Then
$$Indirect^{k}\left(u\right)=\sum_{v\in V_{T,u}}\frac{Local^{k}\left(v\right)}{dist_{T}\left(u,v\right)}$$
with $V_{T,u}$ the set of vertices of the subtree of *T* rooted in *u* and $dist_{T}\left(y,v\right)$ the (graph theory) distance between *u* and *v* in *T*. Note that $Indirect^{k}\left(u\right)$ is a measure of the (bad) impact of the chosen position of *u* (when dealing with player (u,v)) on the positions of $N_{G}\left(u\right)$.

The cost function of each player (u,v) at the end of round *k* of the game is $R^{k}\left(u,v\right)=\frac{Direct^{k}\left(u\right)+Indirect^{k}\left(u\right)}{2}$. The purpose of each player (u,v) is to minimize R(u,v) as much as possible.

Note that the choice of players as couples of vertices (x,y) rather than single vertices x∈V aims to reduce the number of strategies per player (42 here), which has an impact on the efficiency of the method to reach an equilibrium (see next section). Indeed, in the case where a player is a vertex, we should define $42^{\delta_{T}\left(x\right)}$ strategies for each player, where $\delta_{T}\left(x\right)$ represents the out-degree (i.e., the number of children) of vertex *x* in *T*. It is also the reason for all couples (u,v) for any given *u* to have the same utility function value at each round *k* of the game.

### 3.2. Distributed Reinforcement Learning

A Linear Reward Inaction distributed reinforcement learning strategy as defined in [66] is applied in order to make the game reach, *if possible*, a Nash equilibrium. For each round *k* of the game, we hence associate a stochastic vector $V_{x,y}^{k}$ to each (x,y) player, such that $V_{x,y}^{k}\left(s\right)$ is the probability for each strategy s∈Start(x) being a spatial position in I(x). When a player (x,y) has to play, a strategy $s_{x,y}^{k}$ is randomly chosen using the probability $V_{x,y}^{k}$.

At the end of iteration *k*, we define the utility of player (x,y) at round *k* by
$$U^{k}\left(x,y\right)=\frac{Max\left(s_{x,y}^{k}\right)-R^{k}\left(x,y\right)}{Max\left(s_{x,y}^{k}\right)-Min\left(s_{x,y}^{k}\right)}$$
where
$$Max\left(s_{x,y}^{k}\right)=\max_{\begin{array}{c} k^{\prime }\leq k:\\ s_{x,y}^{k^{\prime }}=s_{x,y}^{k} \end{array}}R^{k^{\prime }}\left(x,y\right)\;and\;Min\left(s_{x,y}^{k}\right)=\min_{\begin{array}{c} k^{\prime }\leq k:\\ s_{x,y}^{k^{\prime }}=s_{x,y}^{k} \end{array}}R^{k^{\prime }}\left(x,y\right)$$

Then, the vectors $V_{x,y}^{k+1}\left(s\right)$ are updated considering the Linear Reward Inaction approach of [67], defined as follows:$$V_{x,y}^{k+1}\left(s\right)=V_{x,y}^{k}\left(s\right)\;+\;\left(1-V_{x,y}^{k}\left(s\right)\right)\times b\times u^{k}\left(x,y\right)\;if\;\;s=s_{x,y}^{k}$$
$$V_{x,y}^{k+1}\left(s\right)=V_{x,y}^{k}\left(s\right)\;-\;V_{x,y}^{k}\left(s\right)\times b\times u^{k}\left(x,y\right)\;\;if\;\;s\neq s_{x,y}^{k}$$
where *b* is a slowing factor taken in [0, 1].

For each round *k* of the game, we define $m^{k}$ as the average of $U^{k}\left(x,y\right)$ on all players (x,y). The game is stopped when $m^{k}$ is not improved after at least *t* consecutive rounds, with *t* a input parameter.

### 3.3. Illustration on a Gas Phase Tri-Peptide: Going from a 2D-MolGraph to the 3D Structure

As a first performance evaluation of this reinforcement learning method, we test on the gas phase tri-peptide molecule $NH_{3}^{+}-Ala_{3}-COOH$ composed of 32 atoms. We run the algorithm over 10 blocks, each block consists of 120,000 iterations. The goal is to be able to predict the 3D structure of the peptide molecule in which there is one hydrogen bond between the $NH_{3}^{+}$ and COOH terminal groups as given by the 2D-MolGraph in Figure 10a.

Figure 10b illustrates the 3D structure that we aim to predict. This 3D structure is indeed known from one snapshot of a DFT-MD trajectory. There exists one hydrogen bond (surrounded in red) between the two extremities of the peptide; the hydrogen (white atom) covalently bonded to the nitrogen (blue atom) is the donor of the H-bond directed towards the acceptor oxygen (red atom) of the COOH terminal group of the peptide. Figure 10c illustrates the best solution found by our developed learning method. Based on the $m^{k}$, our algorithm provides the best solution at the first round (see Table 1). Despite the drawings in Figure 10b,c not using exactly the same orientation angle, we can see a significant resemblance between the target structure and the predicted one. In particular, almost all atoms have their correct covalent bond distances with their neighbors being predicted by the method. The covalent bonds between carbon atoms are obtained slightly longer than the expected ones. The hydrogen bond length found in the predicted solution is too short and is therefore closer to a covalent bond, which explains the O-H bond drawn by the software in Figure 10c. The method can be improved by adjusting the possible positions that one atom can take over the icosahedrons. Moreover, polyhedrons could be used instead of icosahedrons, which would possibly improve the final agreement.

Figure 11a provides the distribution of utility function values for one given block (120,000 iterations).

Two stages are seen. The first stage has a fast increase of the utility function, followed by a “plateau” that starts at block 25,000 and over which the utility values fluctuate between 0.90 and 0.95. Figure 11b represents the time evolution of the utility averaged over the best iteration for each block. For this example, one can see that the best solution is reached after 26,768 blocks.

Table 1 provides a statistical view of the evolution of the predicted 3D structure. For each run, the best solution, as described above using the utility function, is selected, for which the RMSD descriptor (Root Mean Square Displacement) between the ‘real/target’ 3D structure and the solution found by our algorithm is calculated. The best solutions in terms of the average utility function $m^{k}$ and RMSD are indicated in green and the worst ones in red. We notice that, even if the best $m^{k}$ is found at the first block, the best prediction in terms of the actual 3D structure is found at the second block with an RMSD equal to 1.551. Other tests led to similar results.

The conclusions that can be extracted from this first application are summarized as follows. (1) The utility function can certainly be improved. One idea would be to redefine both $Direct^{k}\left(u\right)$ and $Indirect^{k}\left(u\right)$ functions by e.g., changing the way the distances are calculated. (2) A factor related to the global structure that is predicted should be added to the utility function in order to improve the prediction. (3) Though the RMSD metric is simple to calculate and is an easy descriptor, it is by no means the best one, as already discussed in the literature. This metric is especially too sensitive to small changes in the 3D structure, and it is therefore hard to use in order to quantify the agreement with the target structure. Another metric has to be found for comparing the predicted 3D structure to the target one.

## 4. Conclusions and Perspectives

As illustrated by the contributions presented in this paper, modeling the structure of molecules and more complex heterogeneous solid–liquid interfaces by topological 2D graphs and algorithmic graph theory is powerful in order to dissect molecular dynamics trajectories into time-dependent conformational analyses and statistical properties. The 2D-MolGraphs that we have developed and presented are versatile and transferable between various phases of matter (from molecules in the gas phase to the more complex condensed matter illustrated here with heterogeneous aqueous interfaces). The graph algorithms applied to the topological 2D graphs were shown to be powerful tools for analyzing the conformational space and were shown to provide conclusions that would be otherwise hard to obtain. Ongoing works in our group aim at applying the 2D-MolGraphs to biomolecules such as proteins. The 2D-MolGraphs and their associated canonical graphs are fast to compute. For instance, on a simple desktop machine (Macbook Pro, i7, 8Go RAM), it takes 0.9 s for one snapshot of BPTI (Bovine Pancreatic Trypsin Inhibitor, 1QLQ, 896 atoms) and 146 s for one snapshot of GLP-1 (Human Glucagon-like Peptide-1 Receptor, 7S15, 6458 atoms). This opens the route to rapid analyses of the trajectories of biomolecules with the 2D-MolGraphs.

The GaTewAY software [42] has been developed and includes the definitions of the 2D-MolGraphs presented in this paper, as well as the graph algorithms applied in this paper. As discussed in the text, this software is easily modified and can be expanded in order to include supplementary definitions for the edges of the 2D-MolGraphs.

We also illustrated the possibility to go beyond the molecular level of granularity of the 2D-MolGraphs (with atoms being the vertices) and hence adopt a more coarse-grained level of representation of a given molecule (with e.g., H-Bonded rings being the vertices of the graph). We have shown that this representation is highly promising to characterize the time evolution of H-Bonded molecules such as peptides. This level of granularity has already been addressed in [23], and it can be used for either predicting the 3D coarse-grained structure of molecules or for searching stable polymorphic substructures in a sequence of conformers (identified by the algorithms of [40]). Some of our ongoing works aim at developing more algorithmic approaches in this topic.

The second part of this paper reported our preliminary works for back-mapping, i.e., predicting a 3D structure of a molecule from a 2D-MolGraph using game theory and reinforcement learning. Our leading idea is to apply stochastic learning methods to find the molecule’s equilibrium space corresponding to stable or metastable situations. This assumes that the 3D structure of a molecule is based on the notion of equilibrium between local components. The first results are promising and show the necessity to include more descriptors related to the global predicted structure in the utility function of the reinforcement learning method. One also has to go beyond the RMSD metric in order to evaluate the relevance of the predicted 3D structures. These are ongoing works. 

## Figures and Tables

**Figure 1 molecules-28-02892-f001:**
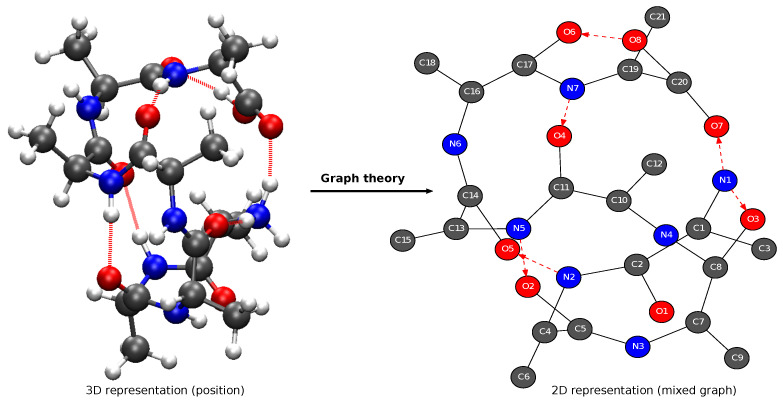
One snapshot (3D representation, (**left**)) and its associated 2D MolGraph (**right**) from a trajectory of $NH_{3}^{+}-Ala_{7}-COOH$ ($C_{21}H_{38}N_{7}O_{8}$) peptide in the gas phase. In the left panel, carbon atoms are represented in dark gray, nitrogen atoms in blue, oxygen atoms in red, and hydrogen atoms in light gray. In the 2D MolGraph (**right**), the conventions are the following: vertices are colored in dark gray, blue, red and light gray, corresponding respectively to a carbon atom, nitrogen atom, oxygen atom and hydrogen atom; the edges are colored and represented in black lines when a covalent bond exists between two atoms (i.e., between two vertices) and in red dashed lines when a hydrogen bond exists between a heavy atom and the H-bond acceptor (i.e., between the associated vertices), which is an arc as seen by the arrow going from donor to acceptor of the H-bond.

**Figure 2 molecules-28-02892-f002:**
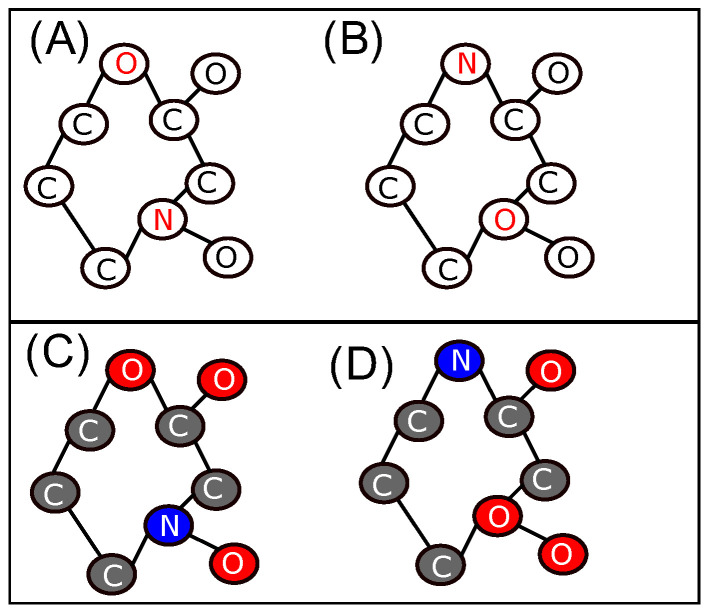
Example of isomorphic conformations represented by their 2D-MolGraphs. Top: conformations (**A**) (left) and (**B**) (right) are isomorphic by interchanging vertices *N* and *O* shown in red regardless of the chemical type of their corresponding atoms. Bottom: conformations (**C**,**D**) are not isomorphic anymore once the coloring process has been applied.

**Figure 3 molecules-28-02892-f003:**
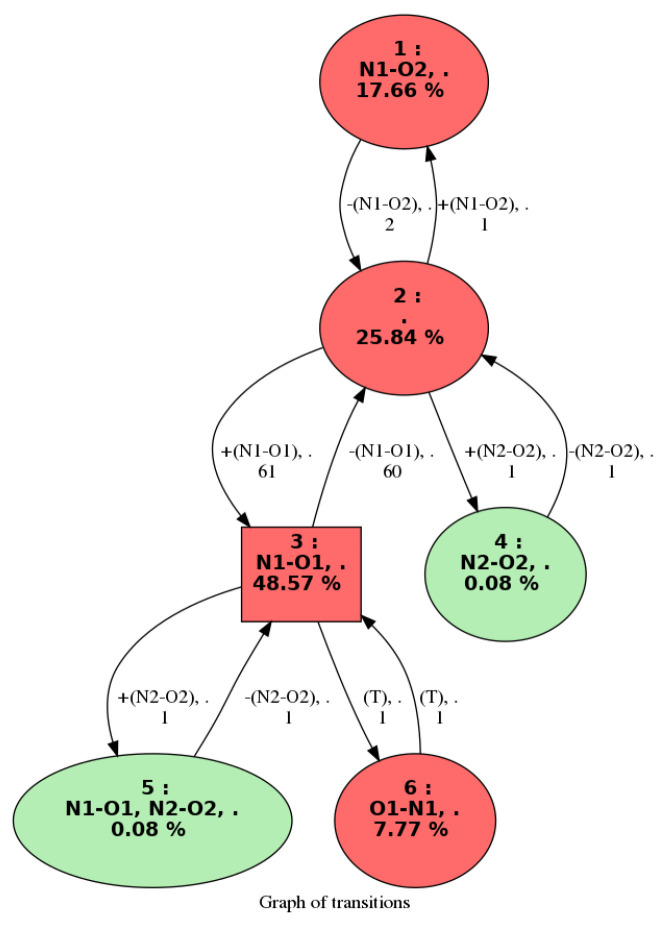
Example of a graph of transitions for a trajectory of $NH_{3}^{+}-Ala_{2}-COOH$ ($C_{6}H_{13}N_{2}O_{3}$) peptide in the gas phase. See text for the nomenclature and colors.

**Figure 4 molecules-28-02892-f004:**
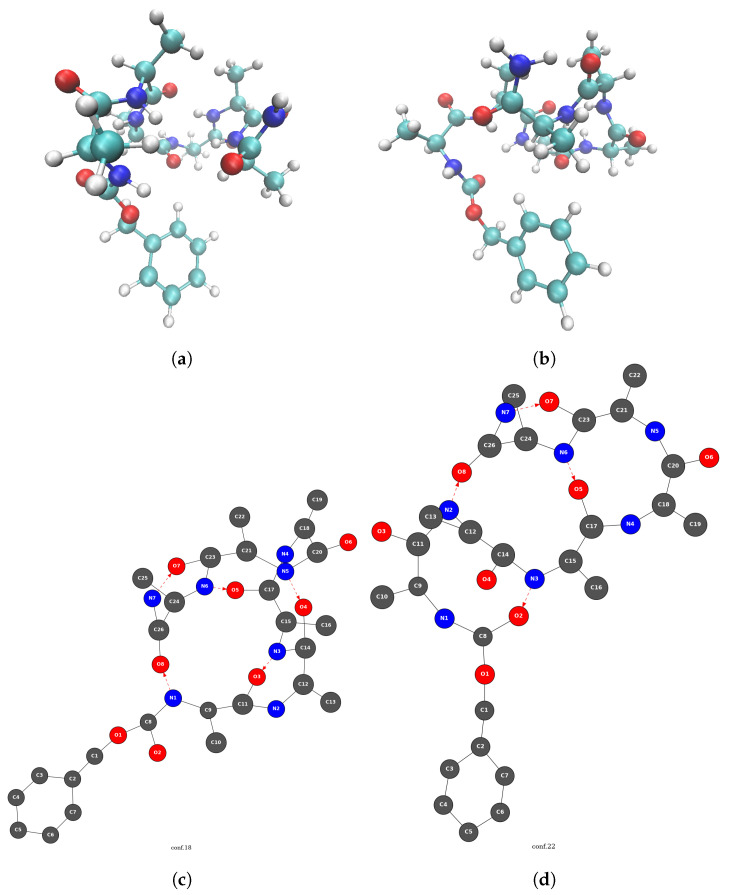
Illustration of two conformers (3D representation, top (**a**,**b**)) and their associated 2D-MolGraphs (bottom (**c**,**d**)) identified in a trajectory of the gas phase $Z-Ala_{6}-COOH$ ($C_{26}H_{39}N_{7}O_{8}$) peptide. Colors for the 3D structures vertices: green for carbon, dark blue for nitrogen, red for oxygen, and light gray for hydrogen atoms. Colors for the 2D-MolGraph vertices: dark gray for carbon, dark blue for nitrogen, and red for oxygen. Solid black edges in the graphs represent covalent bonds, and the red arcs represent the hydrogen bonds directed from the donor to the acceptor.

**Figure 5 molecules-28-02892-f005:**
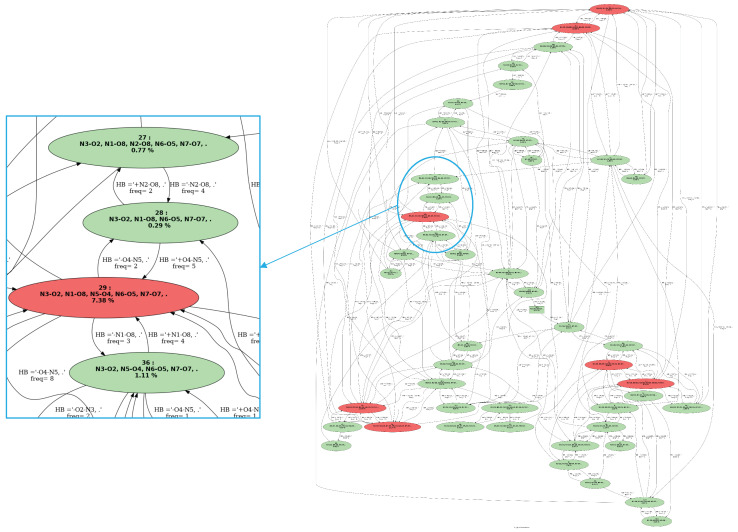
Graph of transitions (**right**) for 150 K dynamics of the gas phase $Z-Ala_{6}-COOH$ peptide. The vertices represent the explored conformers. Vertices in red are for conformers with a total percentage of appearance $P_{min}\%$ greater than 4%, vertices in green are for conformers with $P_{min}\%<4\%$. These threshold values can be modified at will. Edges between vertices represent the transitions between two conformers as observed over time.The labels on each edge provide the total percentage of occurrence of the transition and the associated chemical changes that occur. Edges are directed. **The left figure** is the image zoomed over a portion of the graph of transitions displaying the knowledge on four conformers being explored over time. One conformer appears over 29% of the trajectory (red vertex), and the three other conformers appear for less than the minimum threshold value of time ($P_{min}$) (green vertices). The transitions between these conformations are shown with the black arrows/edges.

**Figure 6 molecules-28-02892-f006:**
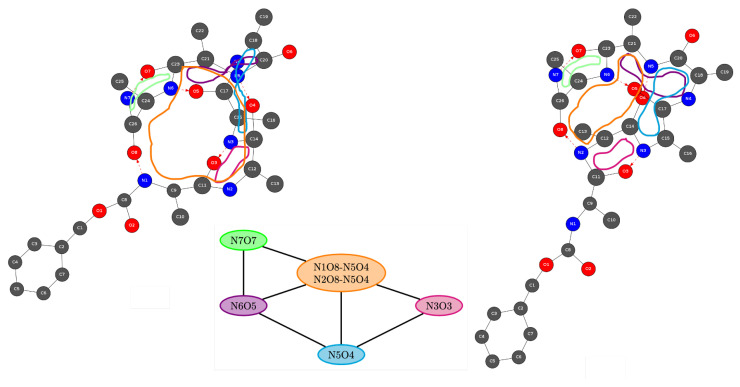
Example of two 2D-MolGraphs of two conformers of gas phase $Z-Ala_{6}-COOH$ ($C_{26}H_{39}N_{7}O_{8}$) peptide. The five H-bonded rings/cycles that exist in each conformation have been highlighted in different colors (orange, green, purple, light blue, pink). The bottom graph is the coarse-grained graph of cycles that is identical for the two conformers. The colors used for the vertices of the graph of cycles are identical to the colors used in the 2D-MolGraphs for representing the H-bonded cycles. See text for more details on the graph of cycles, in particular for the definitions of the edges.

**Figure 7 molecules-28-02892-f007:**
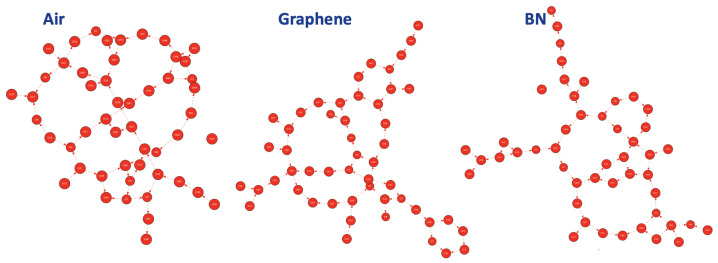
Illustration of a 2D-MolGraph applied to condensed phase systems, for each hydrophobic aqueous interface investigated here: air/liquid water (**left**), graphene/liquid water (**middle**), BN/liquid water (**right**). Only the water molecules located in the BIL (Binding Interfacial Layer; see definitions in [17]) are taken into account for the 2D-MolGraph analysis (∼48 water molecules on average). Vertices of the graph represent the oxygen atoms of the water molecules (red); the dashed red arcs represent the H-bonds between two water molecules oriented from donor to acceptor.

**Figure 8 molecules-28-02892-f008:**
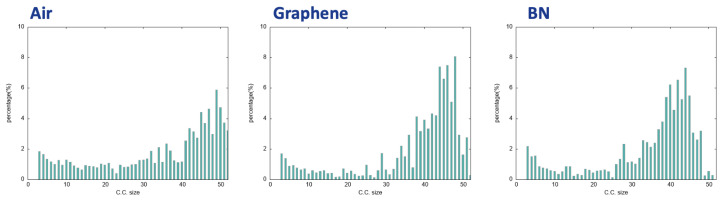
Distribution of the connected components of the 2D-MolGraphs (condensed phase, see text for details) for the air/liquid water interface (**left**), the Graphene/liquid water (**middle**), and the BN/liquid water (**right**).

**Figure 9 molecules-28-02892-f009:**
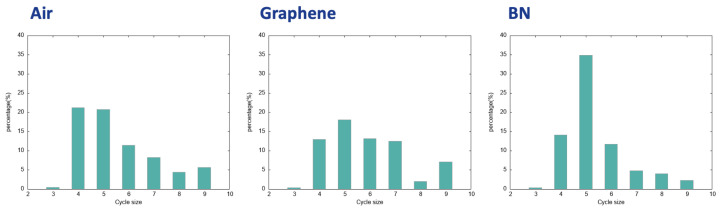
Distribution of the size of H-bonded rings/cycles formed by the water molecules in the 2D-MolGraphs (condensed phase). (**Left**): the air/water interface; (**Middle**): the graphene/water interface; (**Right**): the BN/water interface.

**Figure 10 molecules-28-02892-f010:**
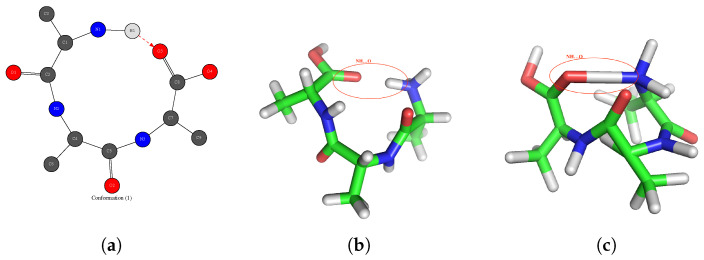
Example for the prediction of the 3D structure of the tri-peptide $NH_{3}^{+}-Ala_{3}-COOH$ from the knowledge of a 2D-MolGraph. (**a**) The 2D-MolGraph of the $NH_{3}^{+}-Ala_{3}-COOH$ ($C_{9}H_{18}N_{3}O_{4}$) tri-peptide extracted from a snapshot of a DFT-MD trajectory. (**b**) Associated 3D structure known from the DFT-MD trajectory. (**c**) The 3D structure of the best predicted solution by the presently developed game theory coupled to the learning method.

**Figure 11 molecules-28-02892-f011:**
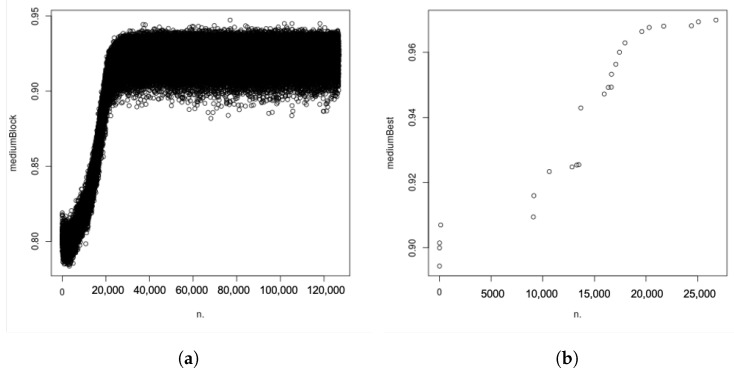
Time evolution of the average of the values of the utility function. (**a**) Time evolution of the average of the utility function values over one block. (**b**) Time evolution for the best iterations at each block.

**Table 1 molecules-28-02892-t001:** Evolution of the solutions found at each block. Green values indicate the best solutions in terms of the average utility function $m^{k}$ and RMSD and the red values represent the worst ones.

Block	1	2	3	4	5	6	7	8	9	10
Best $m^{k}$	0.969	0.955	0.961	0.963	0.953	0.955	0.958	0.962	0.959	0.967
RMSD	2.301	1.551	1.981	2.089	2.200	2.271	2.186	2.071	2.053	1.870

## Data Availability

Data can be made available upon request to the authors.

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
