# Peer review of "Algorithmic Graph Theory, Reinforcement Learning and Game Theory in MD Simulations: From 3D Structures to Topological 2D-Molecular Graphs (2D-MolGraphs) and Vice Versa"

_molecules, 2023, doi:10.3390/molecules28072892_

Round 1

Reviewer 1 Report

The review article by a Bougueroua et al is an interesting trad covering their work on 2D-molecular graph based trajectory analysis and prediction algorithm. The authors have also included their exploratory research for 3D-structure prediction via a game theory type extension. I believe the manuscript can be published as is. 

As a simple comment, it may help readers to follow more easily if the authors note the gas phase and/or condensed phase snapshots in their figures legends.

Reviewer 2 Report

Bougueroua et al. review methods developed in their group to use graph based methods to analyse MD simulation data. They also describe attempts at back-mapping from topological graphs to Cartesian coordinates/3D structure.

The paper is generally well written and I found it fairly accessible, despite not being expert in graph theory. Some of the figures could be improved, however.
Figure 1 nicely shows how 3D structure is mapped to a 2D-MolGraph, and I would have liked to see something similar for Figure 4 as there are aspects of the graph that I cannot intuit from the expected structure. Most of the figures have very small font sizes in places too.

There have been a number of recent efforts using machine learning to map between 3D structure and lower dimensional representations of larger proteins. I expected to see a few of these studies mentioned in the discussion of challenges regarding back mapping, and some justification for the author's choice of approach here.

I also wondered how well the author's methods scale. Is it practical to use their approach to analyse an MD simulation of a typical protein for example?

Reviewer 3 Report

The authors present applications of their graph theory based methods and software to characterize complex structures obtained from molecular simulations. These applications include both structural characterizations and also information on the transition pathways between different structures. This is an interesting work and should help raise awareness in the molecular simulation community of graph theory based methodology in analysing the results of simulations. In this respect, I can recommend the manuscript for publication  in Entropy subject to minor revisions listed below.

-The authors should spell out MolGraphs in the title. Only standard abbreviations should be used.

-The level of the manuscript is accessible, but to reach readers trained in chemistry, it could be useful to translate more of the information/description to the parallel chemical structures.

-Some spelling and grammar points should be corrected. For example “Theroy”, “cristallographic", …

-To show the utility of the graph in representing connectivity of the structures, can the authors show the two polypeptide chemical structures next to their graphs in Fig. 4

-While Fig. 5 is meant to be illustrative and not studies in detail, the details are too small for the interested reader to be able to discern then and get an indication of how the model is working.

-In part 3 of the paper where the authors discuss developing 3D models from 2D graphs, it is stated that the VSEPR model is used as a basis for the start of the extraction of a 3D structure from a 2D graph. Could this not have been done through the chemical knowledge of the bonding and hybridisation of the vertices, as determined partly by the order and color of the vertex of the graph?

- There will be some ambiguity in converting a 2D graph with hydrogen atoms removed to a 3D structure. How is this resolved? As a very simple example, for the simple C-C graph, how does the model distinguish whether the 3D structure is CH3-CH3, CH2=CH2, or CHºCH? This ambiguity may be present in fragments of larger graphs as well.   
